# Effects of Bismuth Exposure on the Human Kidney—A Systematic Review

**DOI:** 10.3390/antibiotics11121741

**Published:** 2022-12-02

**Authors:** Lauter E. Pelepenko, Ana Cristina P. Janini, Brenda P. F. A. Gomes, Adriana de-Jesus-Soares, Marina A. Marciano

**Affiliations:** Department of Restorative Dentistry, School of Dentistry of Piracicaba, State University of Campinas, Av. Limeira, 901, Areião, Piracicaba 13414-903, SP, Brazil

**Keywords:** bismuth, chelation, colloidal bismuth subcitrate, hemodialysis, kidney, renal

## Abstract

The effects of bismuth toxicity on the kidney—the main organ responsible for blood filtration—were systematically reviewed. This review was motivated by availability of several sources of bismuth in contact with humans including environmental, medications, dental materials, and cosmetics, potentially leading to kidney filtration of this chemical. No previous studies have systematically reviewed the literature considering this association. A total of 22 studies with a total of 46 individuals met the inclusion criteria, 19 being case reports with only one patient enrolled. The included studies publication dates ranged from 1961 to 2021 and the countries of publication were the United States of America, United Kingdom, Germany, Turkey, Switzerland, and Canada. Bismuth sources affecting the kidneys were uniquely reported as from medical purposes and mostly associated to overdoses with several symptoms, apparently with dose-dependent consequences. Patient history of renal impairment seemed to affect the outcome of the case. Several therapies were conducted following bismuth intoxication, and few studies performed renal biopsies describing its histological findings. It is crucial to reconsider the nephrotoxicity of bismuth compounds, mainly in patients with previous history of renal impairment.

## 1. Introduction

The kidney is the main organ responsible for blood filtration and excretion of xenophobic substances [1], the nephron being its functional unit [2,3]. The nephron endowment is limited, and the number of nephrons varies widely between individuals in a tenfold magnitude [4,5]. Kidney pathologies include acute kidney injury [6], between 7 and 90 days, this injury is termed acute kidney disease [7]; and chronic kidney disease can be defined by a decreased kidney function of at least 3 months duration shown by glomerular filtration rate deficit, and/or markers of kidney damage, despite the underlying cause [8]. Diagnosis of these renal conditions include several criteria (i.e., serum creatinine, glomerular filtration rate, albuminuria, and urine output and sediment analysis) [9,10,11,12].

Bismuth (from German, “Wismut”, “white matter”) is a chemical element with symbol Bi, atomic number 83 (83 protons and 83 electrons), with an atomic mass of 208.9 u, found in group 15 of the periodic classification of chemical elements. The differentiation from lead (confused by its similarity) of this metal was demonstrated in 1753 by Claude Geoffroy Junine [13]. At room temperature, bismuth is in a solid state. This element is heavy, crystalline, is the most diamagnetic (magnetically repelled) of all metals, and possesses the lowest thermal conductivity of all elements—except for mercury. Of all the metals, bismuth conducts the least electricity [14,15]. Bismuth compounds have been used therapeutically for over two centuries under several names and synonyms; most of its salts have fallen into disfavor—due to its adverse effects—culminating in delisting from drug-regulatory authorities and narrowing its availability to few compounds [16]. The US Food and Drug Administration (FDA) “https://www.fda.gov/search?s=bismuth” (accessed on 12 November 2022) and the European Medicines Agency (EMA) “https://www.ema.europa.eu/en/search/search?search_api_views_fulltext=bismuth” (accessed on 12 November 2022) currently provide drug lists, reports, and guidelines regarding bismuth compounds.

Human exposure to bismuth has been widely reported in studies associating this element with environmental sources [17,18,19,20,21,22,23,24,25,26,27,28,29,30,31,32,33,34], bismuth salts [35,36,37,38,39,40], dental materials [41,42,43] and cosmetics [44,45]. However, the long-term effects of bismuth exposure considering multiple sources is yet to be investigated, as no studies are currently available. Moreover, considering the high blood filtration performed by the kidneys, it is important to review potential effects of bismuth on this organ. Hence, the review question was: Does bismuth exert any effects, conditions, or diseases in humans’ kidneys?

This systematic review aimed to investigate previously reported associations of bismuth exposure with renal consequences in humans.

## 2. Results

The PRISMA flow diagram is shown in Figure 1 with the final full text inclusion of 22 studies. Data extracted from the included studies are provided in Table 1 in chronological order of publication year. The complete list of the 19 excluded studies along with the reasons for their exclusion is shown in Appendix A.

The included studies were from years 1961 to 2021 and the countries of publication were the United States of America, United Kingdom, Germany, Turkey, Switzerland, and Canada. Most of the included studies were case reports with a total of 46 individuals enrolled. Bismuth sources considered were uniquely for medical purposes and mostly associated with overdoses. The associated signs and symptoms to the exposure were several and are described in Table 1, along with the treatment performed for each case followed by its outcome. Two case reports were manually included due to its relevance for the reviewed subject [46,47].

The risk of bias assessment of the included studies is shown in Table 2.

**Table 1 antibiotics-11-01741-t001:** Data extracted from the included studies.

Year	Author	Country	Study	*n*	Age	Gender	Bismuth Source	Associated Signs and Symptoms	Treatment and Outcome
1961	Gryboski et al. [48]	United States of America	case report	1	7 years old	male	30 mg of bismuth thioglycolate (Thio-Bismol) intramuscularly (1.3 mg/kg)	Vomit, mild tenesmus, false urge to void, weight loss, mild periorbital oedema, and azotemia (for creatinine and urea). Later, skin rash was also observed.	Diagnosis of renal damage due to bismuth was based on retention of waste products, low urinary output, inability to concentrate urine, and mild acidosis. Normal renal function was restored within a month.
1963	Chamberlain and Franks [49]	United States of America	case report	1	9 years old	female	3 g of bismuth thioglycolate (Thio-Bismol) intramuscularly	Nausea, vomit, emesis, proteinuria, glycosuria, haematuria, pyuria, periorbital oedema, reduction in urinary output	Diagnosis of acute nephritis. After 9 days, the patient was discharged from the hospital.
1964	Czerwinski and Ginn [50]	United States of America	case report	1	19 years old	female	21 tablets of bismuth sodium trithioglycollamate (Bistrimate) equivalent to 1.5 g of elemental bismuth	Nausea, vomit, hand tingling and numbness, cramping of the legs and thighs, urine glucosuria, proteinuria and granular casts. Azotemia and urine output decrease was also reported. Later, skin rash was also observed.	Renal damage suffered by the patient was due to sodium bismuth trithioglycollamate. After 35 days, the patient was discharged from the hospital.
1968	James [51]	United States of America	case report	1	14 years old	female	7 to 8 tablets of bismuth sodium triglycollamate (Bistrimate) equivalent to 525 to 600 mg of elemental bismuth	Vomit, prostration and lethargy, urine output decrease, azotaemia, mouth dryness, feet numbness, and weight loss. No bismuth was detected in a specimen of serum submitted two days after admission.	There was no history of any pre-existing renal disease. After 10 days, the patient was discharged from the hospital.
1989	Hudson and Mowat [52]	United Kingdom	case report	1	27 years old	male	100 tablets of colloidal bismuth (De-Nol) a total of 12 g of the drug	Anorexia, nausea, vomiting, general malaise, weakness of legs, blurring of vision, thirst, poor urinary output, dehydration, proximal leg muscle weakness with hyperreflexia and ankle clonus. An abdominal X-ray film showed opacification of the colon by ingested bismuth. Bismuth in blood was detected in a decreasing concentration up to 96 days after ingestion, measured by atomic absorption spectrophotometry (AAS).	Renal failure and neurotoxicity induced by bismuth were diagnosed. Five days later, renal function had returned to normal and neurological signs resolved.
1990	Taylor and Klenerman [53]	United Kingdom	case report	1	76 years old	male	80 tablets of tripotassium dicitrato bismuthate (De-Nol)	Vomit, oliguria, haematochezia, and mild epigastric tenderness in admission. Bismuth was detectable in the blood with a concentration of 1600 µg/L. X-ray showed opacification of the colon by ingested bismuth.	Acute abdominal pain developed with absent bowel sounds but he was judged unfit for surgery and died 4 days later. Necropsy revealed a perforated duodenal ulcer and pale kidneys which proved to contain bismuth with a mass fraction of 16 mg/g.
1990	Playford et al. [54]	United Kingdom	case report	1	68 years old	male	For two months, a 10 mL daily dose of tripotassium dicitrato bismuthate (De-Nol) being 864 mg of bismuth per day, a total of 48 g of bismuth.	Confusion, difficulty walking, urinary incontinence, hallucinations, and ataxia. Alterations on the electroencephalogram were consistent with a metabolic encephalopathy. Blood bismuth was 880 µg/L and urinary 230 µg/L detected by inductively coupled plasma source spectrometry (ICP-MS).	This patient had a history of renal impairment. For the bismuth intoxication, a chelation therapy used 2-3 dimercapto-l-propane sulphonic acid and after discontinuation of the bismuth administration its blood concentration decreased and improvement in his cerebral function was observed.
1991	Treiber et al. [55]	Germany	prospective	18 patients: 7 with impaired renal function (creatinine clearance = 34 ± 19 mL/min), 7 with normal renal function, and 4 dialysed. All patients had positive *Helicobacter pylori* status and diagnosis of duodenal ulcers, gastric ulcers, or gastritis type B confirmed by endoscopy.	53.9 ± 14.4 years old for impaired renal function, 44.9 ± 9.2 years old for normal renal function, and 49.0 ± 14.7 years old for dialysed	9 male and 9 female	240 mg of tripotassium dicitrate bismuthate (Telen) twice a day before meals (corresponding to 216 mg bismuth) for four weeks except for the dialysis patients where a 120 mg dose was used.	A rapid increase in the plasma concentration of bismuth was observed in all patients following the first dose when measured by AAS. This increase was dependent on renal function assessed by creatinine levels. All patients noted black stools. Two patients reported nausea and one patient sleep disturbance. One patient reported allergic reaction and another one osteoarthritic symptom.	Monotherapy with tripotassium dicitrate bismuthate resulted in eradication of *Helicobacter pylori* in 28% of the patients. After stopping the therapeutic regimen, the plasma concentrations of bismuth returned to pre-study levels within two weeks (in normal renal function patients) or four weeks (impaired renal function patients). Plasma concentrations in the dialysed patients receiving half of the dose did not exceed those of all the other individuals.
1992	Huwez et al. [56]	United Kingdom	case report	1	21 years old	male	39 tablets of bismuth subcitrate (brand not stated) a total of 4.68 g of the drug	Epigastric pain, reduced urinary output, and deteriorated renal function. Renal biopsy revealed moderate acute tubular necrosis with focally prominent regenerative atypia. Serum bismuth was above 100 µg/L on day 14 after ingestion detected by ICP-MS.	Serum bismuth concentration reduced due to distribution. Patient was discharged well on day 120.
1995	Stevens et al. [46]	United Kingdom	case report	1	21 years old	male	50–60 De-Nol tablets	Nausea, vomiting, diarrhoea, exhaustion, anuria, and azotaemia. Measured by AAS, blood bismuth concentration was 590 µg/L. An abdominal radiograph showed opacification of the colon with bismuth. Evidence for proximal tubular damage in the pathogenesis of the renal failure was shown by glycosuria and a β2-microglobulin excretion.	150 mg of dimercaprol was administered intramuscularly and patient was transferred. Dialysis using a cellulose acetate kidney and a recirculating dialysis system for four hours was commenced one hour after intravenous sodium-(2,3)-dimercaptopropane-(l)-sulphonate administration. The patient began to pass urine again on day 8 and by day 13 was independent of dialysis.
1996	Akpolat et al. [47]	Turkey	case report	1	16 years old	female	10–15 tablets of tripotassium dicitrato bismuthate (brand not stated)	Nausea, vomiting, dizziness, and oliguria. Renal biopsy was performed. Hematoxylin-eosin sections revealed vacuolation, flattening, necrosis, and prominent regeneration in tubular epithelium (mainly proximal tubuli epithelium) with widening of lumens and localized cell detachment. Moderate mononuclear cell infiltration in the interstitium and edema, congestion, and narrowing of Bowman’s capsular space in the glomeruli was also observed.	The final diagnosis was acute tubular necrosis. The patient was managed with haemodialysis via femoral vein catheter. Her urine volume gradually increased, and azotemia subsided. After 20 days, the patient was discharged from the hospital.
2001	İşlek et al. [57]	Turkey	case report	1	2 years old	male	28 tablets of colloidal bismuth subcitrate (De-Nol) a total of 8.4 g of the drug	Vomit and three tablets were observed in the gastric aspirate. X-ray showed opacification of the intestine and the colon by ingested bismuth. Still somnolent and lethargic days later. Reduced urine output, facial edema, and azotemia developed. No bowel movement during the first three days after admission was observed and the stool was black in color on day four.	Acute poisoning due to colloidal bismuth subcitrate was diagnosed and the patient was given gastric lavage in the paediatric emergency service. Intraperitoneal swan-neck catheter was placed for automated peritoneal dialysis therapy for six days. Urine volume gradually increased. A control abdominal X-ray demonstrated no opacification on day six. After 20 days, the patient was discharged from the hospital.
2002	Sarikaya et al. [58]	Turkey	case report	1	17 years old	female	25 tablets of bismuth subcitrate (brand not stated) a total of 7.5 g of the drug	Decreasing urine output and anuric for the last three days. Pretibial and periorbital oedema were observed. Gastric pain, eosinophiluria, azotemia, and increased renal parenchymal echo. Renal biopsy was performed. Signs of acute tubular necrosis were encountered in the proximal tubule epithelium. Epithelial flattening, lumen widening, and atrophic changes were seen in the convoluted tubules and mononuclear cell infiltration and edema in the interstitium.	The patient was managed with hemodialysis via a subclavian vein catheter. Urine volume gradually increased. Urinary examination did not show any protein, erythrocytes, or leukocytes. After 21 days, the patient was discharged from the hospital.
2002	Hruz et al. [59]	Switzerland	case report	1	22 years old	female	5.4 g of colloidal bismuth subcitrate (brand not stated)	Drowsiness, pain over both renal flanks, oliguric, azotemia, proximal tubular dysfunction (Fanconi’s syndrome: hypophosphatemia, hypouricemia, metabolic acidosis, and renal glucosuria despite normal plasma glucose concentration), proteinuria. Measured by AAS, serum bismuth concentration reached 640 µg/L on day three. Ultrasonography showed enlargement of both kidneys. Magnetic resonance imaging showed enlarged and edematous kidneys with thinning of the cortical area. On day 8, the patient showed ulcerations of both tonsils and stomatitis.	Intravenous treatment with the chelating agent sodium- 2,3-dimercapto-1-propanesulfonate was started. Because of persistent renal failure and with the aim to eliminate bismuth, hemodialysis was started. Diuresis resumed 10 days after admission. Signs of Fanconi’s syndrome with tubular proteinuria and the ulcerations on both tonsils disappeared. After 19 days, the patient was discharged from the hospital.
2005	Cengiz et al. [60]	Turkey	case report	1	16 years old	female	60 tablets of colloidal bismuth subcitrate (De-Nol) a total of 18 g of the drug	Nausea, vomiting, facial paraesthesia, and periorbital and pretibial edema. Ultrasonography demonstrated slightly increased kidney size bilaterally and slightly increased echogenicity in the renal parenchyma. Measured by AAS, serum bismuth concentration was 495 µg/L after 12 days of ingestion.	Hemodialysis therapy was started and oral treatment with a metal chelating agent (penicillamine 20 mg/kg per day) was also prescribed. Azotemia subsided after a month. After 16 days, the patient was discharged from the hospital.
2013	Erden et al. [61]	Turkey	case report	1	21 years old	female	20 tablets of colloidal bismuth subcitrate (brand not stated). Each tablet included 300 mg of drug, which is equivalent to 120 mg of bismuth oxide.	Abdominal ultrasonography demonstrated slightly increased echogenicity in the renal parenchyma. Oliguria followed by anuria was observed. Blood chemistry and urine sediment showed signs of proximal tubular dysfunction (Fanconi’s syndrome) and azotemia.	Oral treatment with the chelating agent sodium-2,3-dimercapto-1-propanesulfonate was initiated. Hemodialysis was performed because of anuria and severe metabolic acidosis. Afterwards, the patient’s urine output progressively increased. After 15 days, the patient was discharged from the hospital. Azotemia did not subside, and the patient needed hemodialysis for approximately one year after due to the chronic renal failure caused by the bismuth ingestion.
2014	Kratochwil et al. [62]	Germany	cohort	8 patients: 7 with progressive advanced neuroendocrine liver metastases refractory to treatment and 1 with bone marrow carcinosis	not given	4 male and 4 female	213Bi-DOTATOC receptor-targeted alpha-radionuclide therapy was administered in increasing activities in cycles every 2 months. To minimize renal toxicity, 1000 mL of a nephroprotective solution containing 30 g lysine and 30 g arginine and 500 mL of gelofusine was administered simultaneously, followed by 1000 mL of saline.	Acute haematological toxicity was low. One patient with a previous history of grade IV thrombopenia with 90Y-DOTATOC therapy developed grade II thrombopenia with 213Bi-DOTATOC. During the long-term follow-up, three patients developed chronic anaemia. Erythropoietin levels in lower normal range were found in two patients. Graves’ disease (toxic diffuse goiter) was diagnosed one year after the last treatment cycle and must be considered therapy-associated because thyroid cells can also express somatostatin receptors.	Anemia was interpreted as a side effect of endocrine kidney function rather than bone marrow function. Acute hematological and chronic kidney toxicity are most relevant in peptide receptor radiation therapy. During the follow-up of two years, none of the patients developed severe kidney failure requiring dialysis.
2015	Akinci et al. [63]	Turkey	case report	1	16 years old	female	19 grams of bismuth subcitrate potassium (De-Nol)	Opacity in the upright abdominal X-ray. On the 3rd day of admission, the patient developed acute renal failure, metabolic acidosis, and oliguria. The patient had sore throat from the 3rd day and bilateral tonsillar ulceration was observed. The patient became anuric on the 5th day. On the 15th day of admission the patient developed altered mental state. Neurological examination revealed confusion, somnolence, and cortical blindness, as well as bilateral pyramidal findings followed by frequent seizure attacks. Magnetic resonance imaging showed hyper-intense signal alterations at the levels of bilateral parietal vertices of both cerebellar hemispheres.	Gastric lavage was performed. The patient underwent hemodialysis following catheterization through the jugular vein. Following dialysis performed on alternate days, urination resumed on the 10th day. On the 13th day, the patient became polyuric. Toxic metabolic encephalopathy was associated with the images. The laboratory parameters of the patient began to normalize on the 20th day of admission. After 24 days, the patient was discharged from the hospital.
2017	Disel et al. [64]	Turkey	case report	1	34 years old	female	8 tablets of bismuth subcitrate (De-Nol) a total of 2400 mg of the drug	Nausea, vomit, apathy, proteinuria, glucosuria, and hemoglobinuria. A blue-black discoloration in the teeth and gum was observed. Ultrasonography revealed bilateral perirenal free fluid and grade 1–2 renal parenchymal changes. The patient became anuric and acidotic.	The previous medical history described continuous bismuth therapy for peptic ulcer. Gastric lavage was performed. A diagnosis of acute renal failure due to bismuth toxicity was concluded. The patient underwent hemodialysis and one session of plasmapheresis. Apathy improved and the discoloration of the teeth and gum recovered. After 24 days, the patient was discharged from the hospital.
2019	Çelebi-Tayfur et al. [65]	Turkey	case report	1 pregnant patient	16 years old	female	20 tablets of colloidal bismuth subcitrate (De-Nol). Each tablet included 300 mg of drug, which is equivalent to 120 mg of bismuth oxide	Nausea, vomit, and anxiety. Beta human chorionic gonadotropin level in the blood was 150.07 mIU/mL. Ultrasonography demonstrated severely increased echogenicity on both renal parenchyma. Transvaginal ultrasonography detected an early pregnancy at 5 weeks and 4 days. Proteinuria, hematuria with red blood cells of normal morphology, and pyuria were also observed. The patient rapidly became oliguric, and azotemia was detected.	Decontamination of bismuth was performed by gastric lavage and fluid therapy. A chelation therapy with parenteral dimercaprol was started. Continuous venovenous hemodiafiltration was performed for three days. The patient’s urine output progressively increased and renal function tests comprising arterial blood gas analysis gradually improved. After 21 days, the patient was discharged from the hospital. The council recommended termination of the pregnancy due to potential risks, but the patient preferred continuation of the pregnancy. The patient gave birth to a term healthy baby boy with vaginal delivery.
2020	Halani and Wu [66]	Canada	case report	1	79 years old	male	1 to 2 bottles of bismuth subsalicylate (Pepto-Bismol) daily for 6 months, equivalent of 8.3 to 16.6 g of bismuth subsalicylate per day	Confusion, inattention, impaired hearing, and ataxia. His medical history included chronic kidney disease secondary to diabetic nephropathy.	Acute kidney injury was diagnosed, and Pepto-Bismol was discontinued. Isotonic intravenous fluids were administered to facilitate renal excretion of the salicylate. The mental status, hearing, and gait improved.
2021	Aga et al. [67]	Canada	case report	1	68 years old	male	Self-treated with over-the-counter bismuth subsalicylate in a dose of <150 mg/kg (brand not stated)	The patient was minimally responsive (Glasgow Coma Score 7). Nausea, vomit, metabolic acidosis, and increased colostomy outputs were observed. A prerenal acute kidney injury from emesis and high colostomy outputs was present.	The acute salicylate toxicity after bismuth subsalicylate ingestion successfully managed with hemodialysis with complete neurological recovery.

## 3. Discussion

The review question ‘Does bismuth exert any effects, conditions, or diseases in humans’ kidneys?’ could be partially answered since the present systematic review successfully included 22 studies reporting effects of bismuth exposure associated with the human kidney. However, other sources of bismuth—besides the ones reviewed here—can potentially be in contact with humans without studies investigating its potential effects on the kidneys.

There were two included reports in the 1960s reporting bismuth being used as an injectable solution to treat warts [48,49] in pediatric patients with several renal associated symptoms. In the same decade, two other reports [50,51] associated to bismuth sodium trithioglycollamate tablets taken in overdose described similar symptoms requiring a hospital length of stay from 9 to 35 days. In the 1970s, 942 cases of bismuth intoxication were reported in France, 72 of which were fatal. Currently, this medication is controlled in this country, and its sales condition is reviewed every two years [68].

Chronologically, in the 1980s, the next included report [52] described a case in which the patient had ingested 100 tablets of colloidal bismuth 10 days before looking for medical support with several symptoms. An abdominal X-ray could show opacification of the colon by bismuth ingestion associated with a high concentration of bismuth in blood being detected up to 96 days after the ingestion. It was also reported that the reduction in blood bismuth concentration after hemodialysis was transient since the bismuth seemed to be re-established from tissue stores into the blood.

In 1990, an included study [53] reported the death of the patient after ingestion of 80 tablets of tripotassium dicitrato bismuthate. The necropsy revealed a perforated duodenal ulcer and pale kidneys which proved to contain bismuth with a mass fraction of 16 mg/g. Bismuth toxicity is mainly associated with overdoses, but cases of acute kidney disease associated or not with diabetes, acute hepatitis, osteoporosis, osteomalacia, and chronic encephalopathy have also been reported to prolonged use of bismuth salts [69,70,71,72].

The prolonged use of tripotassium dicitrato bismuthate for 2 months was reported in 1990 [54]. This dosage was associated with confusion, difficulty in walking, urinary incontinence, hallucinations, and ataxia; moreover, alterations on the electroencephalogram were consistent with a metabolic encephalopathy. The bismuth blood concentration detected by inductively coupled plasma source spectrometry (ICP-MS) was elevated and the patient had a history of renal impairment which could be associated with these symptoms. In this case report, the chelation therapy used 2-3 dimercapto-l-propane sulphonic acid and after discontinuation of the bismuth administration, the blood concentration decreased and improvement in the patient cerebral function was observed. The only prospective study included in the present systematic review took place in 1991 [55] and included 18 patients with and without previous renal conditions and prescribed bismuth therapy for *Helicobacter pylori* administered twice a day for four weeks. A rapid increase in the plasma concentration of bismuth was observed in all patients following the first dose, this increase being dependent on renal function assessed by the creatinine levels. The authors also reported that the monotherapy with tripotassium dicitrate bismuthate resulted in eradication of *Helicobacter pylori* in 28% of the patients.

A 120-day case management was reported in 1992 [56] including a patient renal biopsy showing moderate acute tubular necrosis four days after ingestion of 39 tablets of bismuth subcitrate. The authors suggested that hemodialysis can be necessary in acute renal failure cases but had little effect on the rate of elimination from the intracellular compartment in which bismuth seemed to have a long half-life. However, a similar case report of bismuth intoxication from 1995 [46] suggested that chelation associated with hemodialysis (one hour after intravenous sodium-(2,3)-dimercaptopropane-(l)-sulphonate administration) could potentially be useful in such cases.

After renal biopsy, a case report [47] where 10–15 tablets of tripotassium dicitrato bismuthate was ingested, hematoxylin-eosin staining was performed. Vacuolation, flattening, necrosis, and prominent regeneration in tubular epithelium (mainly proximal tubuli epithelium) with widening of lumens and localized cell detachment was observed. Moreover, moderate mononuclear cell infiltration in the interstitium and edema, congestion, and narrowing of Bowman’s capsular space in the glomeruli were also described. These histological findings were reported as compatible with acute tubular necrosis in this case. Similar histological findings were also reported by another case report [58].

In 2001 a pediatric case report [57] described the management of a 2-year-old who ingested 28 tablets of colloidal bismuth subcitrate. No bowel movement during the first 3 days after admission was observed and the stool was black-colored on day 4. Acute poisoning due to colloidal bismuth subcitrate was diagnosed and the patient was given gastric lavage in the pediatric emergency service followed by intraperitoneal swan-neck catheter for automated peritoneal dialysis therapy for six days. A control abdominal X-ray demonstrated no opacification on day 6. The whole hospital management took 20 days.

Nausea, vomit, gastric pain, facial paresthesia, periorbital edema, oliguria, anuria, and azotemia were symptoms in common to reported cases after oral overdose of bismuth tablets [58,59,60]. Ultrasonography was used intending to demonstrated kidney alterations in cases [59,60] followed by hemodialysis. However, in one case [61] azotemia did not subside and the patient needed hemodialysis for approximately one year after bismuth ingestion due to the resulting chronic renal failure.

Isotopes of bismuth in receptor-targeted alpha-radionuclide therapy (213Bi-DOTATOC) were administered in progressive advanced neuroendocrine liver metastases refractory to treatment and bone marrow carcinosis. The authors considered that acute hematological and chronic kidney toxicity are relevant in peptide receptor radiation therapy and during the follow-up of two years, none of the patients developed severe kidney failure requiring dialysis [62]. This was the only study included in the present systematic review regarding bismuth isotopes and its potential renal effects.

The highest accidental reported dose of bismuth subcitrate potassium ingestion was 19 g [63] and—along with other symptoms—the patient had a sore throat from the 3rd day after ingestion and bilateral tonsillar ulceration. The patient became anuric on the 5th day. On the 15th day of admission the patient developed altered mental state and after neurological examination confusion, somnolence, and cortical blindness was observed. Moreover, bilateral pyramidal findings followed by frequent seizure attacks were also reported. Magnetic resonance imaging showed hyper-intense signal alterations at the levels of bilateral parietal vertices of both cerebellar hemispheres and toxic metabolic encephalopathy could be associated with these images. Several hemodialysis sessions were necessary during the length of stay of 24 days. Nearly a thousand cases of encephalopathy were reported in France in patients taking various bismuth salts [73].

A case report in 2017 [64], described, after the ingestion of 8 tablets of bismuth subcitrate—a total of 2400 mg of the drug, several associated symptoms including a blue-black discoloration in the teeth and patient gums. Diagnosis of acute renal failure due to bismuth toxicity was concluded. The patient underwent hemodialysis and one session of plasmapheresis. Apathy improved and the discoloration of the teeth and gum recovered. Another medical study associated dental discoloration with oral bismuth subsalicylate administration [68]; moreover, dental-related studies [42,43,74] have discussed this dentinal discoloration following the use of bismuth-containing materials and chemical instability of bismuth oxide in such materials reported [75].

A pregnant patient ingested 20 tablets of colloidal bismuth subcitrate and rapidly became oliguric [65]. Transvaginal ultrasonography detected an early pregnancy at 5 weeks and 4 days. Gastric lavage, fluid therapy, chelation therapy, and continuous hemodiafiltration was performed; urine output progressively increased and renal function tests improved. After 21 days, the patient was discharged from the hospital. The local council recommended the termination of the pregnancy due to potential risks, but the patient preferred the continuation of the pregnancy that was carried to term to a healthy baby with vaginal delivery. This was the only included report available with a pregnant patient.

Two cases of long-term use of over-the-counter bismuth subsalicylate were recently reported in Canada [66,67]. In the 2020 case, confusion, inattention, impaired hearing, and ataxia were reported. The medical history included chronic kidney disease secondary to diabetic nephropathy where acute kidney injury was diagnosed, and Pepto-Bismol was discontinued. Isotonic intravenous fluids were administered to facilitate renal excretion of the salicylate. The mental status, hearing, and gait improved. In the 2021 case, the patient was minimally responsive upon admission and the acute toxicity after bismuth subsalicylate ingestion was successfully managed with hemodialysis with neurological recovery.

It seems that chronic bismuth intake can potentially lead to toxicity in the form of encephalopathy, whereas acute toxicity manifests as nephrotoxicity [56,60]. Over-the-counter self-medication can pose an additional risk in managing intoxications [66] and the risk of toxicity should be included in the safety labels for bismuth subsalicylate products, especially regarding patients with previous renal conditions [67]. In addition, it was reported that therapies including the combined intake of multiple daily doses of milk and cream combined with bismuth subcarbonate—intending to alleviate gastric ulcer symptoms—resulted in hypercalcemia, metabolic alkalosis, and cases of acute kidney injury (Milk-alkali syndrome) [76]; a potential association with over-the-counter calcium supplements for osteoporosis to this syndrome was also recently reviewed [77].

## 4. Materials and Methods

The Preferred Reporting Items for Systematic Review and Meta-Analysis (PRISMA) guidelines [78] were used for performing a systematic review of the literature. The International Prospective Register of Systematic Reviews (PROSPERO) registration was obtained previously to the review under the registration number CRD42021270323.

### 4.1. Selection Criteria

The inclusion criteria were when only humans were enrolled and the study mentioned any association between bismuth and kidney effects, conditions, or diseases. No restrictions were applied; however, the meshed search words were used in the English language.

The exclusion criteria were studies not analyzing bismuth or kidney implications not related to bismuth were reported. Moreover, animal studies, in vitro analysis, reviews, editorial comments, and conference abstracts were also excluded.

### 4.2. Search Strategy and Study Selection

The question for the systematic review was: Does bismuth exert any effect, condition, or disease in humans’ kidneys? Searches were performed—on 2 July 2022—using the databases: PubMed, Embase, and gray literature (opengrey.eu). No date of publication nor idiom restrictions were applied.

The words mesh used for all databases was:

(kidney) AND (bismuth)

In sequence, the search results were exported as files compatible with the Mendeley reference manager (Desktop version 1.19.8) where duplicates were removed, and two previously calibrated reviewers assessed the search results according to the inclusion criteria for the ‘full paper’ assessment based initially on study title and abstract. After the initial screening, the selected studies were reassessed now with the full copies available for the final step of the study inclusion. Disagreements in any of the described steps were resolved by a third person.

### 4.3. Data Extraction

A spreadsheet (Excel Microsoft 365 version 2110) was previously developed for data extraction including: year of publication, country, type of study, population enrolled, age, gender, bismuth source, associated signs and symptoms, and reported treatment and outcome. The same reviewers performed the data extraction and inconclusive decisions were discussed with a third person.

### 4.4. Risk of Bias Evaluation in Individual Studies

The risk of bias evaluation used a questionary tool [79] to perform this analysis based on the eight questions of the evaluation instrument.

## 5. Conclusions

It can be concluded that bismuth can be toxic for humans. Most of the studies included in the present systematic review were case reports with one patient only and the effects of the bismuth intoxications on the kidneys seemed to be dose dependent. It is crucial to reconsider the nephrotoxicity of bismuth compounds, especially in patients with previous history of renal impairment.

## Figures and Tables

**Figure 1 antibiotics-11-01741-f001:**
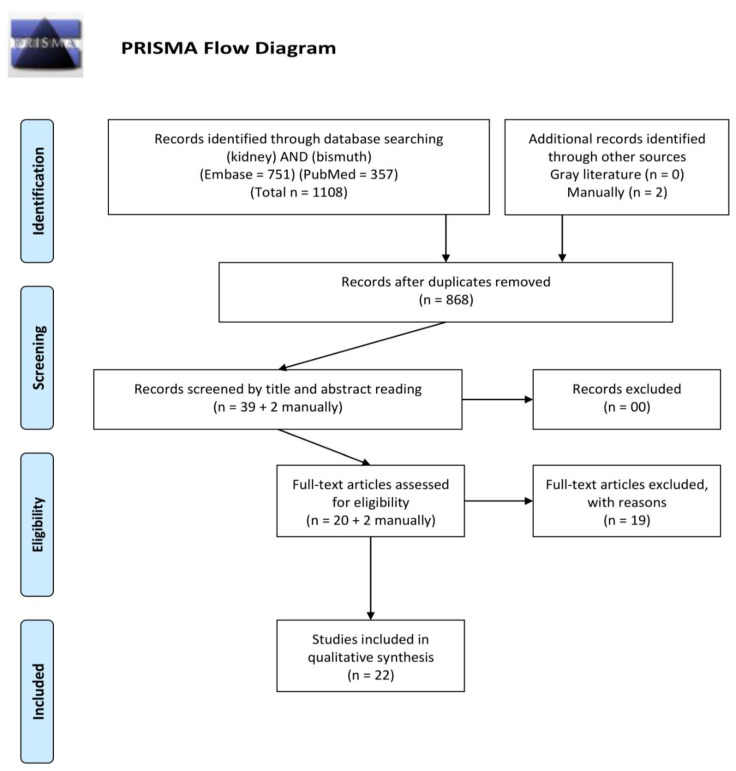
PRISMA flow diagram.

**Table 2 antibiotics-11-01741-t002:** Risk of bias assessment of the included studies.

Year	Author	Q1	Q2	Q3	Q4	Q5	Q6	Q7	Q8
1961	Gryboski et al. [48]	Unclear	Yes	Yes	Yes	Yes	Yes	No	Yes
1963	Chamberlain and Franks [49]	Unclear	Yes	Yes	Yes	Yes	Unclear	No	Unclear
1964	Czerwinski and Ginn [50]	Yes	Yes	Yes	Yes	Yes	Yes	No	Unclear
1968	James [51]	Yes	Yes	Yes	Yes	Yes	Yes	No	Yes
1989	Hudson and Mowat [52]	Yes	Yes	Yes	Yes	Yes	Yes	No	Unclear
1990	Taylor and Klenerman [53]	Yes	Yes	Yes	Yes	Yes	Yes	No	Yes
1990	Playford et al. [54]	Yes	Yes	Yes	Yes	Yes	Yes	No	Yes
1991	Treiber et al. [55]	Not applicable	Not applicable	Not applicable	Not applicable	Not applicable	Not applicable	Not applicable	Not applicable
1992	Huwez et al. [56]	Yes	Yes	Yes	Yes	Yes	Yes	No	Yes
1995	Stevens et al. [46]	Yes	Yes	Yes	Yes	Yes	Yes	No	Yes
1996	Akpolat et al. [47]	Yes	Yes	Yes	Yes	Yes	Yes	No	Yes
2001	İşlek et al. [57]	Yes	Yes	Yes	Yes	Yes	Yes	No	Yes
2002	Sarikaya et al. [58]	Yes	Yes	Yes	Yes	Yes	Yes	No	Yes
2002	Hruz et al. [59]	Yes	Yes	Yes	Yes	Yes	Yes	No	Yes
2005	Cengiz et al. [60]	Yes	Yes	Yes	Yes	Yes	Yes	No	Yes
2013	Erden et al. [61]	Yes	Yes	Yes	Yes	Yes	Yes	No	Yes
2014	Kratochwil et al. [62]	Not applicable	Not applicable	Not applicable	Not applicable	Not applicable	Not applicable	Not applicable	Not applicable
2015	Akinci et al. [63]	Yes	Yes	Yes	Yes	Yes	Yes	No	Yes
2017	Disel et al. [64]	Yes	Yes	Yes	Yes	Yes	Yes	No	Yes
2019	Çelebi-Tayfur et al. [65]	Yes	Yes	Yes	Yes	Yes	Yes	No	Yes
2020	Halani and Wu [66]	Yes	Yes	Yes	Yes	Yes	Yes	No	Yes
2021	Aga et al. [67]	Yes	Yes	Yes	Yes	Yes	Yes	No	Yes

Q1 Were patient’s demographic characteristics clearly described? Q2 Was the patient’s history clearly described and presented as a timeline? Q3 Was the current clinical condition of the patient on presentation clearly described? Q4 Were diagnostic tests or assessment methods and the results clearly described? Q5 Was the intervention(s) or treatment procedure(s) clearly described? Q6 Was the post-intervention clinical condition clearly described? Q7 Were adverse events (harms) or unanticipated events identified and described? Q8 Does the case report provide takeaway lessons?

## Data Availability

PROSPERO registration was obtained under the code: CRD42021270323.

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
