# Peer review of "Effects of Bismuth Exposure on the Human Kidney—A Systematic Review"

_antibiotics, 2022, doi:10.3390/antibiotics11121741_

Round 1

Reviewer 1 Report

Dear Authors,

congratulations on taking up such a broad topic that has been well developed.

The tables  are the perfect complement to this text.

When working on such a large material, it is not difficult to find a few inaccuracies.

Introduction :

general information is missing

- History of bismuth

- Properties of bismuth - what compounds does it create?

- table with listed drugs that contain bismuth

 - chemical - graphical representation of the element bismuth - 

https://www.britannica.com/science/bismuth

Discussion :

It is worth reminding the reader of such a team

-Milk-Alkali Syndrome; Rimsha Ali   Chinmay Patel In: StatPearls [Internet]. Treasure Island (FL): StatPearls Publishing; 2022 Jan. 2022 May 15. 

Calcium-Alkali Syndrome: Historical Review, Pathophysiology and Post-Modern Update.  Zayed RF, Millhouse PW, Kamyab F, Ortiz JF, Atoot A.Cureus. 2021 Feb 11;13(2):e13291

Conclusions :

I miss a few sentences about the toxicity of bismuth for a living organism/human

References :

-It is really  difficult to browse literature if the item begins with the initials of the author's name. I suggest starting with the author's surname, then the first name in literature

Author Response

Dear reviewer, 

A response letter was uploaded.

Thank you very much.

Reviewer 2 Report

Systematic review titled (Effects of bismuth toxicity on the human kidney – a systematic review) by Pelepenko et al. discusses the impact and toxic effect of bismuth on the kidney. I think the topic does not  fit the scope of "Antibiotics". 

1- Introduction: first paargraph:; is too preliminary & first part of it  is like a student lecture & does not fit to a scitific publication.

2- Abstract should be better explained & amended by som numerical values

3- Title: does not fit the content, I think authors mean exposure to bismuth & nephrotoxicity 

4- References are very old

5- Tables are very very large & desgined in non-publishable manner.

6- In general, ENglish use is not perfect, authors should be consistent in utilizing terms throughout the article & to be more precise in describing their intentions.

Author Response

(The authors gave the same response as above.)

Reviewer 3 Report

Manuscript Title “Effects of bismuth toxicity on the human kidney – a systematic 2 review” has been well written and concise. However, few below points can be added to make it further better.

Line 30: “when lasting between seven and ninety days” Author should make changes “between seven to ninety days”

Line 34: Diagnosis of these renal conditions include several criteria: Author should include details of criteria involved in diagnosis.

As per author in line no. 17: few studies performed renal biopsies describing histological findings; Inclusion of histological diagrams will add advantage to the manuscript and establish more relevance of the review.

Line 184: Necropsy image addition would be more relevant and give an insight and connect

Section 4, Line no. 289, 298, 313, 320; should be sub-numbered.

Author must include separate section containing current ways of Bismuth toxicity and how it can be prevented.

Author Response

(The authors gave the same response as above.)

Round 2

Reviewer 2 Report

I recommend accepting the current form of this paper in Antibiotics.

thanks